# Can LLMs Handle WebShell Detection? Overcoming Detection Challenges with Behavioral Function-Aware Framework

**Feijiang Han**
University of Pennsylvania
feijhan@seas.upenn.edu

**Jiaming Zhang & Chuyi Deng**
Central South University
{8213200215,8208220112}@csu.edu.cn

**Jianheng Tang**[*] **& Yunhuai Liu**[*]
Peking University
{tangentheng@stu.pku, yunhuai.liu@pku}.edu.cn

## Abstract

WebShell attacks, where malicious scripts are injected into web servers, pose a significant cybersecurity threat. Traditional machine learning and deep learning methods are often hampered by challenges such as the need for extensive training data, catastrophic forgetting, and poor generalization. Recently, Large Language Models (LLMs) have emerged as a powerful alternative for code-related tasks, but their potential in WebShell detection remains underexplored. In this paper, we make two major contributions: (1) a comprehensive evaluation of seven LLMs, including GPT-4, LLaMA 3.1 70B, and Qwen 2.5 variants, benchmarked against traditional sequence- and graph-based methods using a dataset of 26.59K PHP scripts, and (2) the Behavioral Function-Aware Detection (BFAD) framework, designed to address the specific challenges of applying LLMs to this domain. Our framework integrates three components: a Critical Function Filter that isolates malicious PHP function calls, a Context-Aware Code Extraction strategy that captures the most behaviorally indicative code segments, and Weighted Behavioral Function Profiling (WBFP) that enhances in-context learning by prioritizing the most relevant demonstrations based on discriminative function-level profiles. Our results show that, stemming from their distinct analytical strategies, larger LLMs achieve near-perfect precision but lower recall, while smaller models exhibit the opposite trade-off. However, all baseline models lag behind previous State-Of-The-Art (SOTA) methods. With the application of BFAD, the performance of all LLMs improves significantly, yielding an average F1 score increase of 13.82%. Notably, larger models like GPT-4, LLaMA-3.1-70B, and Qwen-2.5-Coder-14B now outperform SOTA benchmarks, while smaller models such as Qwen-2.5-Coder-3B achieve performance competitive with traditional methods. This work is the first to explore the feasibility and limitations of LLMs for WebShell detection and provides solutions to address the challenges in this task.

## 1 Introduction

The rapid growth of web applications and the expansion of cloud-based services have intensified the cybersecurity threat landscape, with WebShells emerging as a significant concern. WebShells are malicious scripts injected into web servers that allow attackers to remotely execute arbitrary commands, steal sensitive data, and compromise system integrity (Starov et al., 2016; Cui et al., 2018). According to a Cisco Talos report (Talos, 2024), threat groups used a variety of web shells against vulnerable or unpatched web applications in 35% of incidents in Q4 2024, a sharp increase from the previous quarter, when such activity was observed in only 10% of incidents. These attacks are particularly dangerous

---

[*]Corresponding authors

due to the stealthy nature of WebShells, which are rapidly evolving to evade traditional detection methods (Hannousse & Yahiouche, 2021).

In response to this challenge, the community has proposed several approaches. Rule-based methods that rely on predefined signatures or heuristics are increasingly ineffective against the complexity and diversity of modern WebShells (Le et al., 2021; Jinping et al., 2020). Machine learning models, especially deep learning techniques (Pu et al., 2022), have shown promise in addressing these threats. However, they require extensive training on large datasets–resources that are often difficult to obtain and sensitive in nature (Shang et al., 2024). In addition, these models face challenges such as catastrophic forgetting and poor generalization, especially when dealing with obfuscated or encrypted attacks (Jinping et al., 2020; Zhang et al., 2025).

Recently, Large Language Models (LLMs) have gained attention for a variety of code-related tasks, including code generation (Ma et al., 2024) and vulnerability detection (Liu & He, 2023; Wang et al., 2025). Studies have shown that with effective prompt engineering, LLMs can perform remarkably well without additional training (Nong et al., 2024; Trad & Chehab, 2025). They also provide interpretable explanations for their decisions, which is critical in cybersecurity contexts (Ma et al., 2024). Despite these advantages, there has been limited exploration of LLMs for WebShell detection.

Detecting WebShells with LLMs presents unique challenges that differ from other code analysis tasks. WebShells often employ obfuscation and encryption techniques and are embedded in large codebases dominated by benign content (Liu & He, 2023). For example, the largest WebShell in our dataset spans 1,386,438 tokens, far exceeding the context window of most LLMs, which risks truncating critical malicious segments when processing entire source files (Wang et al., 2025; Ceka et al., 2024). In addition, in-context learning (ICL) struggles in this domain: the variability and obfuscation of WebShells complicate the selection of effective demonstrations, and these examples further occupy significant context space, reducing capacity for the target code (Yuan et al., 2024). While recent research has focused on increasing the context length of LLMs (Chen et al., 2023), studies suggest that the performance of an LLM tends to degrade with longer inputs, and the low processing speed may also become unacceptable for practical use (Ma et al., 2024; Fang et al., 2024).

**In this paper, we present two key contributions to advance the application of LLMs in WebShell detection.**

**First,** we systematically evaluate LLMs in the context of WebShell detection, comparing their performance with traditional state-of-the-art (SOTA) machine learning methods. Specifically, we evaluate seven closed-source and open-source LLMs of different sizes–including GPT-4 (Achiam et al., 2023), LLaMA 3.1 70B (Grattafiori et al., 2024), Qwen 2.5 Coder (14B/3B) (Yang et al., 2024), and Qwen 2.5 (3B/1.5B/0.5B) (Yang et al., 2024)–on a dataset containing 26.59K PHP scripts (4.93K WebShells and 21.66K benign samples). Our analysis reveals several key findings:

- Larger LLMs, such as GPT-4 and Qwen 2.5 Coder 14B, achieve near-perfect precision but struggle with recall (e.g., GPT-4's recall is 85.98%). Their deep contextual analysis, while powerful, can be deceived by well-disguised malicious code, leading to false negatives.

- Smaller LLMs, such as Qwen 2.5 Coder 3B and Qwen 2.5 0.5B, exhibit high recall but suffer from low precision (e.g., Qwen 2.5 Coder 3B's precision is 38.93%). This stems from their reliance on surface-level pattern matching, which often flags any use of sensitive functions without sufficient contextual validation.

- Randomly selected ICL demonstrations degrade LLM detection performance; examples selected based on semantic similarity to the source code do not yield significant improvements.

For this comparison, we benchmark against several traditional methods, including GloVe+SVM (Petridis, 2024; Rigutini et al., 2024), CodeBERT+Random Forest (Alghamdi et al., 2022), and graph-based approaches such as GAT (Kang et al., 2023). The best-

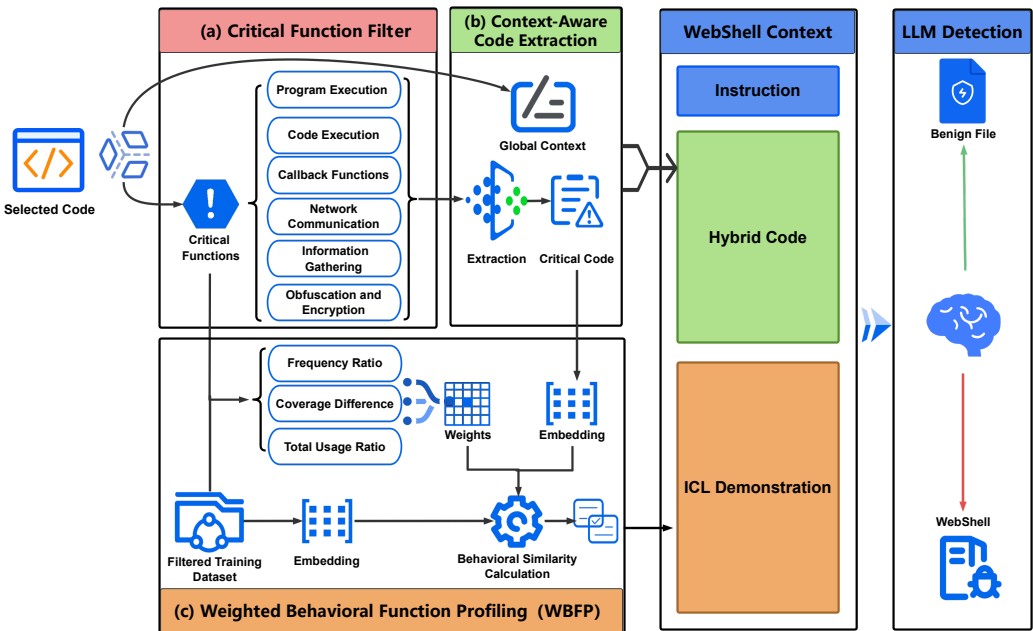

Figure 1: Overview of the Behavioral Function-Aware Detection framework for WebShell detection. It consists of three components: (a) Critical Function Filter, which identifies PHP functions associated with malicious behavior; (b) Context-Aware Code Extraction, which isolates critical code regions to overcome LLM context limitations; and (c) Weighted Behavioral Function Profiling, which selects ICL demonstrations using a behavior-weighted similarity score.

performing LLM, Qwen 2.5 Coder 14B, achieves an F1 score of 96.39%, although it still lags behind GAT-based methods, which achieve an F1 score of 98.87%.[1]

**Second,** we present the Behavioral Function-Aware Detection (BFAD) framework to address the identified challenges and improve the performance of LLMs to meet the requirements of downstream applications by achieving a more balanced trade-off between precision and recall. Our framework combines risk-filtering techniques with an enhanced ICL strategy that uses weighted demonstration selection to prioritize examples most closely related to key malicious behaviors. Experimental results show that our approach improves the average F1 score across all LLMs by 13.82%, with GPT-4 and Qwen 2.5 0.5B improving by 6.89% and 51.23%, respectively. In several cases, such as GPT-4, LLaMA 3.1 70B, Qwen 2.5 Coder 14B, and Qwen 2.5 Coder 3B, our approach enables performance that is competitive with or even superior to traditional methods.

To the best of our knowledge, this is the first work to systematically analyze the feasibility and limitations of applying LLMs to WebShell detection.

## 2 Behavioral Function-Aware Detection Framework

We present the **Behavioral Function-Aware Detection (BFAD)** framework, a comprehensive solution designed to improve WebShell detection by identifying critical code segments and improving the quality of ICL demonstration selection. The architecture of BFAD, as shown

---

[1]While traditional models such as GAT have demonstrated superior performance, they require extensive training and significant data collection efforts. In contrast, LLMs enable direct detection through prompts, leveraging their pre-trained knowledge with minimal task-specific resources to deliver competitive results.

in Figure 1, consists of three primary components: (a) **Critical Function Filter**, which is based on malicious behavior patterns and filters critical PHP function calls to identify key functions; (b) **Context-Aware Code Extraction**, which addresses the limitations of LLMs in handling long input sequences by selectively extracting critical code regions; and (c) **Weighted Behavioral Function Profiling**, which improves demonstration selection by calculating a weighted similarity score based on function-level profiling.

## 2.1 Critical Function Filter

WebShells typically rely on specific PHP functions that facilitate malicious actions such as code execution, data exfiltration, or obfuscation. However, these functions are often embedded in complex, obfuscated code, making detection difficult. To address this, we are developing a **Critical Function Filter** that classifies PHP functions into six different behavioral categories: *Program Execution*, *Code Execution*, *Callback Functions*, *Network Communication*, *Information Gathering*, and *Obfuscation and Encryption*. This taxonomy reflects the different roles these functions play in WebShells.

Specifically, the **Program Execution** category includes functions such as `exec` and `system`, which execute system-level commands and are often exploited in WebShells to execute arbitrary payloads. Similarly, **Code Execution** functions such as `eval` and `preg_replace` interpret input as executable code, allowing attackers to inject arbitrary scripts. The **Callback Functions** category includes functions such as `array_map` and `register_shutdown_function`, which allow dynamic invocation of functions often used to obfuscate malicious code.

In addition, **Network Communication** functions such as `fsockopen` and `curl_init` allow remote communication for data exfiltration and command-and-control operations. **Information Gathering** functions, such as `phpinfo` and `getenv`, are used by attackers to gather system details. Finally, **Obfuscation and Encryption** functions, such as `base64_encode` and `openssl_encrypt`, help disguise or encrypt malicious payloads to avoid detection.

Our statistical analysis (detailed in Appendix C) shows that WebShell files use critical functions far more often than benign files. On average, WebShells contain 22.76 calls to critical functions, compared to only 0.74 in benign files, underscoring their behavioral complexity and distinctiveness.[2]

## 2.2 Context-Aware Code Extraction

Building on the Critical Function Filter, we introduce a **Context-Aware Code Extraction** strategy that identifies and extracts the critical code regions that indicate malicious behavior. These regions focus on the identified critical functions and their surrounding contexts, ensuring that the LLM focuses on the most relevant parts of the code.

The complete extraction procedure is formalized in Algorithm 1, which takes as input the source code $\mathcal{C}$, the list of critical functions $\mathcal{F}$, and the context window size $\tau$, and produces a set of extracted critical code regions $\mathcal{R}$.

We reduce the input size by selectively extracting critical regions of code and merging overlapping segments while preserving behavioral specificity. However, this approach may inadvertently exclude certain global contextual information and increase the emphasis on critical functions, potentially leading to false positives when analyzing benign files that legitimately use such functions. To mitigate this, we append truncated, non-overlapping code segments when context length allows, ensuring that the model receives a balanced representation of local and global code context.

---

[2]While this disparity in critical function usage is significant, it alone does not reliably distinguish WebShells from benign files, since legitimate scripts may also call these functions. Therefore, we use LLMs for deeper contextual analysis to improve detection accuracy.

---

**Algorithm 1** Context-Aware Code Extraction

---
1: **Input:** Source code $\mathcal{C}$, list of critical functions $\mathcal{F}$, context window size $\tau$
2: **Output:** Extracted critical code regions $\mathcal{R}$
3: Initialize empty set of regions: $\mathcal{R} \leftarrow \varnothing$
4: **for** each function $f \in \mathcal{F}$ **do**
5:     Locate all occurrences of $f$ in $\mathcal{C}$
6:     **for** each occurrence of $f$ at position $p$ **do**
7:        Extract context window $[p - \tau, p + \tau]$ from $\mathcal{C}$
8:        Add the extracted region to $\mathcal{R}$
9:     **end for**
10: **end for**
11: Merge overlapping regions in $\mathcal{R}$
12: Compute remaining context budget $B$
13: **if** $B > 0$ **then**
14:     Select additional non-overlapping code segments from $\mathcal{C}$
15:     Add selected segments to $\mathcal{R}$
16: **end if**
17: **return** $\mathcal{R}$

---

## 2.3 Weighted Behavioral Function Profiling

Based on the extracted critical code regions, we propose **Weighted Behavioral Function Profiling (WBFP)**, a method that computes a weighted similarity score to identify behaviorally similar examples for ICL effectively. WBFP assigns weights to each function type based on its prevalence and usage in WebShell versus benign files, quantified by three metrics: coverage difference ($r_c$), frequency ratio ($r_f$), and usage ratio ($r_u$). The coverage difference measures the proportion of files containing a specific function across the two datasets. The frequency ratio is the ratio of the average number of occurrences of the function per file in WebShell files to the average number per file in benign files. The usage ratio reflects the total number of function occurrences in WebShell files compared to those in benign files. These metrics are combined to calculate a discrimination score for each function type

$$\text{Score}_f = (r_c \cdot \alpha) + (r_f \cdot \beta) + (r_u \cdot \gamma),$$

where $\alpha$, $\beta$, and $\gamma$ are empirically determined weights, set to 1 for balanced contribution in our experiments (see Section 3.2). We normalize the discrimination scores to weights:

$$w_f = \frac{\text{Score}_f}{\sum_{f' \in \mathcal{F}} \text{Score}_{f'}}.$$

WBFP then uses the embeddings $E(\cdot)$ generated by *st-codesearch-distilroberta-base* (Abi Akl, 2023; Al-Kaswan et al., 2023) to compute the similarity between the files $x$ and $y$. Let $\mathcal{F}$ denote all critical function types. For each function type $f \in \mathcal{F}$, we concatenate critical regions $R_f(x)$ and compute their embeddings:

$$\mathbf{e}_f(x) = E(\text{concat}_f(x)),$$

The semantic similarity between the files $x$ and $y$ for the function type $f$ is given by

$$s_f(x, y) = \frac{\mathbf{e}_f(x) \cdot \mathbf{e}_f(y)}{\|\mathbf{e}_f(x)\| \|\mathbf{e}_f(y)\|}.$$

The final similarity between the files is the weighted sum of the similarities:

$$\text{Sim}(x, y) = \sum_{f \in \mathcal{F}} w_f \cdot s_f(x, y).$$

This weighted similarity prioritizes function types critical to WebShell behavior, reducing the impact of irrelevant semantic features. Using this score, WBFP ensures that ICL demonstrations accurately capture malicious patterns, improving detection performance.

### 2.4 LLM-Based Detection Framework

We integrate the BFAD framework into the LLM-based detection system, which combines the context-aware code extraction strategy and WBFP to optimize the use of LLMs for WebShell detection. By incorporating both critical code regions and global context, along with behaviorally relevant demonstrations, our framework enhances the ability of the LLM to accurately identify malicious code patterns.

The input to the LLM consists of two main components: (a) a **system directive** that defines the model's role as a cybersecurity expert, and (b) a **user query** that contains the extracted critical code segments and a selected ICL demonstration. To balance efficiency and performance, we limit the user query to one ICL demonstration, which reduces computational overhead while preserving sufficient context for reliable detection. Our prompt is described in detail in Appendix B.

## 3 Experiment

### 3.1 Dataset Overview

We constructed a comprehensive dataset consisting of 21,665 benign PHP programs and 4,929 WebShells. The benign programs were obtained from established open-source PHP projects to ensure applicability to real-world scenarios. The WebShells were collected from public security repositories and augmented with synthetic obfuscation techniques to increase diversity. [3] Using GPT-4's tokenizer, we analyzed the token lengths of both sample types. The WebShell samples had a maximum token length of 1,386,438 and an average of 30,856.60 tokens, compared to a maximum of 305,670 tokens and an average of 2,242.89 tokens for benign programs. These results indicate that WebShells are typically significantly longer than benign samples. A detailed summary of the dataset composition can be found in the Table 3 in the Appendix D.

### 3.2 Experiment Setup

**ICL Settings.** We randomly selected 60% of the dataset to create a fixed, known demonstration library for ICL. Using this subset, we computed normalized scores for different function categories based on the WBFP method, giving equal weight to coverage difference ($r_c$), frequency ratio ($r_f$), and usage ratio ($r_u$) to profile functions according to their behavioral importance in distinguishing WebShells from benign programs. These scores, detailed in the Table 4 in the Appendix D, guided the selection of ICL demonstrations from the known sample library.

**Baseline Models, Hyperparameter Settings, and Evaluation Metrics.** We compared our approach to several baselines: GloVe + SVM, CodeBERT + Random Forest, GCN, and GAT. For GloVe + SVM, we used pre-trained GloVe embeddings with a dimensionality of 300 and an SVM classifier with default parameters (Qi et al., 2018; ZENG et al., 2025). For CodeBERT + Random Forest, we used CodeBERT embeddings with a hidden dimension of 768 and a Random Forest classifier with default settings (Wang et al., 2024a). The GCN was trained with a learning rate of 0.001 over 120 epochs, with 3 hidden layers and a hidden dimension of 32 (Feng et al., 2024). The GAT was trained with a learning rate of 0.001 over 120 epochs,

---

[3]We acknowledge the potential for data leakage due to the pre-training of LLMs. However, the autoregressive nature of this process does not explicitly capture WebShell classification, although it may affect other code generation tasks. To further minimize any risk of leakage, we restricted our selection of PHP programs to projects updated between October 2024 and 2025, thus reducing overlap with LLM training corpora.

with 3 hidden layers, a hidden dimension of 8, and 8 attention heads (Feng et al., 2024). We evaluated the models using standard classification metrics: accuracy, precision, recall, and F1 score.

# 4   Results and Analysis

In this section, we systematically evaluate the performance of LLMs of different scales for WebShell detection and assess the improvements provided by our proposed BFAD framework. Our analysis addresses three research questions (RQs) to explore both the baseline LLM capabilities and the effectiveness of the BFAD components:

- **RQ1:** How do large and small-scale LLMs perform in WebShell detection compared to traditional ML and DL methods, and how does BFAD improve their effectiveness?

- **RQ2:** How effective is context-aware code extraction at balancing global context and local behavioral focus under LLM context length constraints?

- **RQ3:** How does WBFP improve demo selection for ICL?

## 4.1   Performance Evaluation of LLMs and BFAD Enhancements (RQ1)

To answer our first research question, we evaluated seven LLMs against traditional ML and DL baselines. The LLMs included large-scale models (GPT-4, LLaMA-3.1-70B, Qwen-2.5-Coder-14B) and small-scale models (Qwen-2.5-Coder-3B, Qwen-2.5-3B, Qwen-2.5-1.5B, Qwen-2.5-0.5B). Baselines consisted of sequence-based methods (Glove+SVM, CodeBERT+RF) and graph-based methods (GCN, GAT). The detailed performance comparison is presented in Table 1.

Table 1: Performance Comparison of BFAD-Enhanced Models Against Baselines.

| Category | Model | Accuracy | Precision | Recall | F1 Score |
|---|---|---|---|---|---|
| Sequence Baselines | GloVe+SVM | 96.20% | 93.30% | 94.30% | 93.80% |
| | CodeBERT+RF | 96.30% | 94.00% | 95.60% | 94.80% |
| Graph Baselines | GCN | 96.90% | 94.40% | 95.30% | 94.90% |
| | GAT | 98.37% | 99.52% | 97.39% | 98.87% |
| LLM Baselines (Large) | GPT-4 | 97.27% | 100.00% | 85.98% | 92.46% |
| | LLaMA-3.1-70B | 98.01% | 97.31% | 92.36% | 94.77% |
| | Qwen-2.5-Coder-14B | 98.64% | 99.32% | 93.63% | 96.39% |
| LLM Baselines (Small) | Qwen-2.5-Coder-3B | 71.11% | 38.93% | 99.32% | 55.93% |
| | Qwen-2.5-3B | 93.72% | 78.03% | 91.84% | 84.37% |
| | Qwen-2.5-1.5B | 43.62% | 34.61% | 95.77% | 50.84% |
| | Qwen-2.5-0.5B | 19.47% | 18.65% | 100.00% | 31.44% |
| LLM + BFAD | GPT-4 | 99.75% | 100.00% | 98.71% | 99.35% (+6.89) |
| | LLaMA-3.1-70B | 99.38% | 98.72% | 98.09% | 98.40% (+3.63) |
| | Qwen-2.5-Coder-14B | 98.76% | 98.68% | 94.90% | 96.75% (+0.36) |
| | Qwen-2.5-Coder-3B | 78.89% | 46.67% | 100.00% | 63.64% (+7.71) |
| | Qwen-2.5-3B | 97.39% | 88.64% | 99.36% | 93.69% (+9.32) |
| | Qwen-2.5-1.5B | 80.40% | 48.51% | 100.00% | 65.33% (+14.49) |
| | Qwen-2.5-0.5B | 91.94% | 71.10% | 98.73% | 82.67% (+51.23) |

**Performance Summary**   Our experimental results yield two core findings. First, vanilla LLMs exhibit a stark, size-dependent trade-off between precision and recall, with all models lagging behind the state-of-the-art GAT baseline. For instance, GPT-4 achieves 100% precision but only 85.98% recall, while the small Qwen-2.5-0.5B model reaches 100% recall but a mere 18.65% precision. This indicates that out-of-the-box LLMs are not well-calibrated for the nuanced task of WebShell detection, making targeted intervention necessary for effective deployment.

Second, the BFAD framework dramatically improves performance across all models, elevating large LLMs to surpass the GAT baseline and making small models viable competitors. With BFAD, GPT-4's F1 score increases by 6.89 points to 99.35%, exceeding the GAT baseline. The most significant gain is seen in Qwen-2.5-0.5B, whose F1 score improves by a remarkable 51.23 points. This demonstrates that by strategically guiding the model's focus on behaviorally relevant code, BFAD effectively bridges the performance gap, unlocking the potential of LLMs for this task without resource-intensive retraining.

**Analysis of Model Behavior: Misclassification Patterns**  The performance disparities observed in our results appear to stem from fundamental differences in the models' analytical reasoning. The precision-recall imbalance in baseline LLMs is a direct reflection of their reasoning strategy. Small models like Qwen-2.5-0.5B tend to rely on *surface-level heuristics*; they identify keywords like base64_decode and flag them as malicious without assessing the surrounding code's intent. This leads to high recall but generates numerous false positives when such functions are used legitimately (e.g., in a cryptography library). In contrast, large models like LLaMA-3.1-70B employ *deep contextual reasoning*, correctly identifying legitimate use cases and thus achieving high precision. However, this sophistication can be a double-edged sword, as these models can be misled by well-disguised WebShells, rationalizing away malicious indicators and causing false negatives that reduce recall. Our proposed BFAD mitigates these issues by guiding smaller models to better understand function-level behaviors and helping larger models focus on critical functions. As a result, it significantly improves detection performance across scales.

**Analysis of Model Behavior: Counterintuitive Scaling**  We observed a counterintuitive scaling effect where the Qwen-2.5-1.5B model, despite outperforming the 0.5B variant in baseline tests, performed worse with BFAD. We attribute this to differing sensitivities to ICL across scales. The smallest models, like the 0.5B variant, are highly susceptible to label bias and tend to mimic the ICL demonstration's label without deep reasoning (Fei et al., 2023). Because BFAD provides accurately matched demonstrations via WBFP, the 0.5B model benefits disproportionately from this behavior. In contrast, the 1.5B model begins to exhibit more independent reasoning based on its pre-training knowledge. This emergent but still imperfect reasoning can conflict with the ICL example's guidance, degrading performance. This aligns with prior findings that LLMs sometimes rely more on pre-training than on provided examples (Peng et al., 2025). Therefore, the 0.5B model's superior performance with BFAD does not indicate greater intelligence but is an artifact of its strong alignment with the high-quality examples provided.

## 4.2 The Effectiveness of Context-Aware Code Extraction (RQ2)

We evaluated the effectiveness of Context-Aware Code Extraction using two models: GPT-4, a large-scale model, and Qwen-2.5-3B, a smaller-scale model. Three configurations were compared: (1) predictions based on the full source code, (2) predictions using only extracted critical regions, and (3) a hybrid approach combining critical regions with truncated source code. Results are reported in Tables 5 and 6 in Appendix E.

**Impact of Critical Regions**  For the smaller model, Qwen-2.5-3B, critical regions significantly improve performance over the full source baseline. At $\tau = 100$, the F1 score increases from 84.37% to 90.91% (+6.54%), with precision increasing from 78.03% to 86.71% (+8.68%) and recall increasing from 91.84% to 95.54% (+3.70%). This improvement validates the focus on behaviorally relevant code snippets, which reduces irrelevant context and sharpens the focus of the model. However, as the context length ($\tau$) increases, performance decreases - F1 drops to 88.17% at $\tau = 300$, likely due to the model's limited ability to handle extended context, which introduces noise that degrades accuracy.

For the larger model, GPT-4, critical regions increase recall but slightly decrease precision. At $\tau = 300$, recall improves from 85.98% to 96.18% (+10.20%), while precision drops from 100.00% to 98.69% (-1.31%). The F1 score increases from 92.46% to 97.42% (+4.96%), indicating that GPT-4 effectively uses localized behavioral cues to improve recall. However,

the reduced global context may introduce small biases, leading to a precision trade-off, although the overall performance remains strong due to the model's greater capacity.

**Balancing Precision and Recall with the Hybrid Strategy** The hybrid strategy improves model performance by effectively balancing precision and recall. For Qwen-2.5-3B, it increases precision over using critical regions alone, from 86.71% to 89.02% (+2.31%) for $\tau = 100$ and from 82.32% to 85.55% (+3.23%) for $\tau = 300$. Although recall decreases slightly, the F1 score increases to 89.70% at $\tau = 300$ (+1.53% from 88.17%), indicating that the hybrid approach reduces noise in longer contexts and supports smaller models by maintaining focus on critical regions while integrating valuable global context. For GPT-4, the strategy improves recall without compromising precision: at $\tau = 300$, recall increases from 85.98% (using the source code) to 96.82% (+10.84%), while precision remains steady at 100.00%, yielding an F1 score of 98.38%–a 5.92% gain over the source code and a 0.96% improvement over critical regions alone.

## 4.3 The Effectiveness of WBFP for In-Context Learning (RQ3)

We evaluated the effectiveness of WBFP for ICL demonstration selection using Qwen-2.5-3B and GPT-4. This evaluation builds on the optimal hybrid strategy from RQ2. Four demonstration selection strategies were compared: Random Selection (Random), Source Code Semantic Similarity (SC-Sim), WBFP with Equal Weights (WBFP-Eq), and WBFP with Function-Level Weights (WBFP-Wt). Results are detailed in Tables 7 and 8 in Appendix E.

**Limitations of Random and Semantic Similarity-Based Selection** Random selection makes ICL performance much worse by adding irrelevant examples, which makes both models much less effective. For Qwen-2.5-3B, the F1 score drops to 60.83% under the hybrid strategy with a context length of $\tau = 100$, which is a 30.14% reduction from the no-ICL baseline of 90.97%. This is mainly because there is a large drop in precision from 89.02% to 46.33%, although recall remains strong at 88.53%. Similarly, for GPT-4 with $\tau = 300$, the F1 score drops to 76.22%, a 22.16% decrease from the No-ICL baseline of 98.38%, driven by a drop in precision from 100.00% to 65.58%. These results indicate that Random Selection does not provide contextually relevant demonstrations, making it ineffective for WebShell detection.

The SC-Sim approach, which relies on semantic similarity calculated over entire source code samples, also doesn't work well. For Qwen-2.5-3B, SC-Sim achieves an F1 score of 84.36%, a notable decrease from the No-ICL baseline of 90.97%, with a precision of 78.57% and a recall of 91.08%. For GPT-4, it achieves an F1 score of 96.32%, with perfect precision, but a reduced recall of 92.90% compared to the No-ICL baseline's 96.82%. This limited performance is likely due to the dominance of behaviorally irrelevant code segments in the similarity calculation, which dilutes the focus on critical behavioral patterns essential for accurate WebShell identification.

**Superiority of WBFP in Demonstration Selection** For both models, WBFP-Wt consistently outperforms in accuracy, precision, recall, and F1 score, demonstrating its robustness and adaptability for improving ICL in WebShell detection tasks. Specifically, for Qwen-2.5-3B, WBFP-Wt achieves an F1 score of 93.69%, which is 9.33% higher than SC-Sim and 1.28% higher than WBFP-Eq. This is because WBFP-Wt achieves a precision of 88.64% (compared to 78.57% for SC-Sim and 86.39% for WBFP-Eq) and a near-perfect recall of 99.36%. By focusing on the important parts of WebShell behavior, WBFP-Wt makes up for the fact that the smaller model doesn't understand as much. This results in demonstrations that closely match the desired behavioral profiles, improving both precision and recall.

For GPT-4, WBFP-Wt achieves the highest F1 score of 99.35%, which is 3.03% higher than SC-Sim and 0.34% higher than WBFP-Eq . While precision remains at 100.00% across all WBFP variants, WBFP-Wt increases recall to 98.71%, compared to 92.90% for SC-Sim and 98.03% for WBFP-Eq. This improvement highlights WBFP-Wt's ability to leverage GPT-4's advanced understanding of context, matching selected demonstrations to the behavioral characteristics of the target sample to optimize recall without sacrificing precision.

## 5 Related Work

**WebShell Detection Techniques** Early efforts in WebShell detection relied on rule-based methods that used signature matching or heuristics to identify malicious scripts (Le et al., 2021; Jinping et al., 2020). For example, Le et al. (2021) proposed H-DLPMWD, a hybrid approach that combines pattern matching with a CNN to detect ASP.NET WebShells, achieving 98.49% accuracy by using Yara-based filtering and opcode indexing. However, such methods struggle against obfuscated or novel WebShell variants due to their reliance on predefined patterns (Hannousse & Yahiouche, 2021). Machine learning (ML) techniques have advanced this landscape by extracting features from code text or runtime behavior. Jinping et al. (2020) introduced a mixed-model approach using Random Forest and CNNs with N-gram and TF-IDF features that achieved 97% accuracy on PHP WebShells, but it requires balanced datasets and struggles with encrypted samples. Deep learning (DL) has further improved adaptability, with models such as CodeBERT for semantic analysis (Pu et al., 2022) and Graph Attention Networks (GAT) for structural insights (Zhang et al., 2025). (Zhang et al., 2025) proposed MMFDetect, which fuses CodeBERT-CL semantics with CNN-extracted visual features from RGB-mapped PHP code, achieving 99.47% accuracy on evasive WebShells. Despite these gains, ML and DL methods require extensive labeled data-a scarcity in cybersecurity-and exhibit limited generalization to obfuscated threats, along with high computational cost (Shang et al., 2024; Jinping et al., 2020).

**LLMs in Code Analysis** LLMs have revolutionized code-related tasks by leveraging large pre-training corpora for applications such as code generation (Ma et al., 2024), vulnerability detection (Sun et al., 2024), and program reliability assessment (Liu et al., 2024). Ma et al. (2024) demonstrated the ability of LLMs to generate evasive WebShells using hybrid prompts, highlighting their code synthesis potential. Sun et al. (2024) introduced LLM4Vuln, which enhances vulnerability reasoning through knowledge retrieval and has achieved practical success in Solidity audits. These models excel at zero-shot and few-shot learning via prompt engineering (Nong et al., 2024), providing interpretability critical for security contexts (Ma et al., 2024). However, their application to WebShell detection remains under-explored, with previous studies focusing on generation or general vulnerabilities rather than detection of stealthy, context-heavy WebShells.

**Challenges of LLMs for WebShell Detection** Using LLMs for WebShell detection reveals critical bottlenecks. The fixed context window truncates large WebShells, potentially missing malicious segments embedded in benign code (Ceka et al., 2024; Wang et al., 2025). Fang et al. (2024) found that LLM performance degrades with longer inputs, with GPT-4's accuracy dropping to 87% on obfuscated JavaScript. Solutions such as chunking or sparse attention (e.g., LongCoder (Guo et al., 2023), SparseCoder (Wang et al., 2024b)) mitigate this, but often lose global context (Wang et al., 2025). In-context learning (ICL), a cornerstone of LLM adaptability, falters because demonstrations consume context space (Min et al., 2022; Wang et al., 2025), and random or semantic similarity-based selections fail to capture WebShell-specific behaviors. Liu & Wang (2023) proposed maximum information gain for ICL, but its focus on text classification limits its applicability to code security.

These gaps between rule-based rigidity, ML and DL data dependency, and LLM context and ICL limitations motivate our BFAD framework. BFAD overcomes context constraints with a hybrid extraction strategy that preserves critical regions and global cues, outperforming naive LLM applications. In addition, our WBFP enhances ICL by prioritizing behaviorally relevant demonstrations, outperforming generic similarity-based selection methods.

## 6 Conclusion

This paper systematically demonstrates that baseline LLMs are suboptimal for WebShell detection, exhibiting a precision-recall trade-off that underperforms state-of-the-art methods. We propose the Behavioral Function-Aware Detection (BFAD) framework, which overcomes these limitations by focusing analysis on critical code segments and optimizing in-context learning. BFAD enables our LLM-based approach to surpass traditional benchmarks.

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

# A Limitations and Future Work

Our research also highlights several limitations that present important directions for future investigation in this emerging field.

## A.1 Dataset Generalization

A fundamental challenge in WebShell research is the scarcity of diverse, real-world data. Our reliance on public datasets, like that of prior work, limits the evaluation of model generalization against novel, sophisticated threats. To address this, future work should focus on using LLMs to **generate large-scale synthetic benchmarks**. A large-scale, LLM-generated dataset would provide a privacy-preserving and diverse alternative for stress-testing model robustness against a wider range of attack vectors.

## A.2 Model and System Robustness

Our framework's static analysis approach remains vulnerable to advanced obfuscation and zero-day attacks. Future work should therefore focus on enhancing model robustness and system adaptability. To counter evasion, models could integrate **dynamic analysis** by examining runtime function call traces from sandboxed execution. To deepen semantic understanding, **domain-specific fine-tuning** of code-centric LLMs on curated WebShell datasets is a crucial next step. For practical deployment, these enhanced models could be integrated into a **multi-stage "fast and slow" system**, using a smaller model for rapid filtering and a larger one for in-depth analysis of high-risk samples. To ensure long-term resilience, this pipeline could be governed by an **autonomous agent** that continuously identifies new attack patterns and automatically updates the detection models.

# B Prompt Details

---

**Prompt for WebShell Detection**

**System Prompt:** You are tasked with analyzing PHP scripts. Your objective is to classify the provided PHP code as either a WebShell or a legitimate script. A WebShell is typically a malicious script intended to exploit the server by executing unauthorized commands or providing backdoor access.
**User Prompt:** Analyze the provided PHP code to determine whether it constitutes a WebShell or a legitimate script. Provide your verdict as WebShell or benign.
[Critical Code]
[Source Code]
[Examples]
**Output:**

---

## C    Critical Function Details

Table 2: Statistics of Critical Functions in WebShell and Benign Programs. This table reports the percentage of files containing each function category and the average number of occurrences per file, with the "Total" row aggregating statistics across all categories.

| Function Category | Metric | WebShell Files | Normal Files |
|---|---|---|---|
| Program Execution | Files with Function (%) | 53.06 | 1.54 |
| | Avg. Occurrences per File | 3.21 | 0.03 |
| Code Execution | Files with Function (%) | 85.03 | 14.79 |
| | Avg. Occurrences per File | 8.30 | 0.36 |
| Callback Functions | Files with Function (%) | 34.69 | 6.47 |
| | Avg. Occurrences per File | 0.92 | 0.11 |
| Network Communication | Files with Function (%) | 50.34 | 2.77 |
| | Avg. Occurrences per File | 1.69 | 0.04 |
| Information Gathering | Files with Function (%) | 46.26 | 2.77 |
| | Avg. Occurrences per File | 5.46 | 0.05 |
| Obfuscation and Encryption | Files with Function (%) | 69.39 | 9.86 |
| | Avg. Occurrences per File | 3.19 | 0.16 |
| **Total (All Functions)** | Files with Function (%) | 91.16 | 20.49 |
| | Avg. Occurrences per File | 22.76 | 0.74 |

# D  Dataset Details

Table 3: Dataset Composition, Distribution, and Sources. The dataset comprises 26,594 PHP scripts, categorized into benign programs and WebShells, with their respective counts, proportions, and sources.

| Category | Count | Percentage | Source References |
|---|---|---|---|
| Benign Programs | 21,665 | 81.5% | Grav, OctoberCMS, Laravel, WordPress, Joomla, Nextcloud, Symfony, CodeIgniter, Yii2, CakePHP, Intervention/Image, Typecho |
| WebShells | 4,929 | 18.5% | WebShell, WebshellSample, Awsome-Webshell, PHP-Bypass-Collection, WebShell (tdifg), Webshell (lhlsec), PHP-Backdoors, Tennc/Webshell, PHP-Webshells, BlackArch/Webshells, Webshell-Samples, Programe, WebshellDetection, WebShellCollection, PHP-Backdoors (1337r0j4n), PHP-Webshell-Dataset, Xiao-Webshell |
| Total | 26,594 | 100.0% | — |

Table 4: Normalized Scores for Key Function Categories. These scores reflect the weighted behavioral significance of each category as computed by the WBFP method.

| Function Category | Normalized Score |
|---|---|
| Program Execution | 0.2068 |
| Code Execution | 0.2081 |
| Callback Functions | 0.0790 |
| Network Communication | 0.1498 |
| Information Gathering | 0.1861 |
| Obfuscation and Encryption | 0.1702 |

# E   Results

Table 5: Performance of Context-Aware Code Extraction with Qwen-2.5-3B with Different Context Lengths and Strategies.

| Method | Accuracy | Precision | Recall | F1 Score |
|---|---|---|---|---|
| Source Code (Vanilla) | 93.72% | 78.03% | 91.84% | 84.37% |
| Critical Regions ($\tau = 100$) | 96.28% | 86.71% | 95.54% | 90.91% |
| Critical Regions ($\tau = 200$) | 95.78% | 84.75% | 95.54% | 89.82% |
| Critical Regions ($\tau = 300$) | 95.04% | 82.32% | 94.90% | 88.17% |
| **Hybrid Strategy ($\tau = 100$)** | **96.40**% | **89.02**% | **92.99**% | **90.97**% |
| Hybrid Strategy ($\tau = 200$) | 95.66% | 85.47% | 93.63% | 89.36% |
| Hybrid Strategy ($\tau = 300$) | 95.78% | 85.55% | 94.27% | 89.70% |

Table 6: Performance of Context-Aware Code Extraction with GPT-4 with Different Context Lengths and Strategies.

| Method | Accuracy | Precision | Recall | F1 Score |
|---|---|---|---|---|
| Source Code (Vanilla) | 97.27% | 100.00% | 85.98% | 92.46% |
| Critical Regions ($\tau = 100$) | 98.51% | 99.32% | 92.99% | 96.05% |
| Critical Regions ($\tau = 200$) | 99.01% | 99.34% | 95.54% | 97.40% |
| Critical Regions ($\tau = 300$) | 99.01% | 98.69% | 96.18% | 97.42% |
| Hybrid Strategy ($\tau = 100$) | 99.01% | 100.00% | 94.90% | 97.39% |
| Hybrid Strategy ($\tau = 200$) | 99.14% | 100.00% | 95.81% | 97.86% |
| **Hybrid Strategy ($\tau = 300$)** | **99.38**% | **100.00**% | **96.82**% | **98.38**% |

Table 7: Comparison of Demonstration Selection Strategies for In-Context Learning with Qwen-2.5-3B (Under Best Hybrid Strategy $\tau = 100$).

| Method | Accuracy | Precision | Recall | F1 Score |
|---|---|---|---|---|
| No-ICL | 96.40% | 89.02% | 92.99% | 90.97% |
| Random | 77.79% | 46.33% | 88.53% | 60.83% |
| SC-Sim | 93.42% | 78.57% | 91.08% | 84.36% |
| WBFP-Eq | 96.98% | 86.39% | 99.32% | 92.41% |
| **WBFP-Wt** | **97.39%** | **88.64%** | **99.36%** | **93.69%** |

Table 8: Comparison of Demonstration Selection Strategies for In-Context Learning with GPT-4 (Under Best Hybrid Strategy $\tau = 300$).

| Method | Accuracy | Precision | Recall | F1 Score |
|---|---|---|---|---|
| No-ICL | 99.38% | 100.00% | 96.82% | 98.38% |
| Random | 89.00% | 65.58% | 90.97% | 76.22% |
| SC-Sim | 98.63% | 100.00% | 92.90% | 96.32% |
| WBFP-Eq | 99.62% | 100.00% | 98.03% | 99.01% |
| WBFP-Wt | **99.75%** | **100.00%** | **98.71%** | **99.35%** |

