# OpenReview forum: "Can LLMs Handle WebShell Detection? Overcoming Detection Challenges with Behavioral Function-Aware Framework"
_colmweb.org/COLM/2025/Conference — COLM 2025_

### Official Review · Reviewer_EuGo · 2025-04-12

**Rating:** 7
**Confidence:** 3
**Ethics Flag:** 1

**Summary:**

This paper uses Large Language Models (LLMs) to tackle the important and growing threat of WebShell attacks—malicious PHP scripts embedded into web servers.. It introduces a novel framework called Behavioral Function-Aware Detection (BFAD), which enhances LLM-based detection by focusing on malicious function patterns and optimizing context usage through weighted in-context learning. The authors evaluate seven LLMs (including GPT-4, LLaMA 70B, and Qwen variants) on a large dataset (26.6K scripts, ~5K WebShells) and compare them against state-of-the-art traditional detectors.

The empirical evaluation is strong, and the methods are grounded in a principled understanding of both cybersecurity challenges and the limitations of LLMs. The paper demonstrates that with BFAD, LLMs—particularly larger ones—can not only match but outperform the best traditional models. Overall, this is a high-quality, original, and significant contribution that effectively bridges LLM capabilities and real-world cybersecurity needs.

I have a few questions. If they are already mentioned somewhere and I have missed them, please point them out. Otherwise, please let me know what you think.

1. Can BFAD offer interpretability for security analysts? How can we trust the LLM’s reasoning in a practical setting?
2. How well does the model generalize to unseen tactics? Would it require retraining or updates to the critical function list?
3. How do failure cases differ between LLM-based and traditional models? Could a hybrid detection strategy be even better?

Minor comments

1. Instead of “Catastrophic forgetting”, maybe “generalization decay” might be a more accurate term. What do you think?
2. Claims of “near-perfect precision” are correct, but it is worth noting this came with recall trade-offs. Right?
3. In figure 1, the writing in red at top (a) Crital Function Filter --- should be "critical", right?
4. “preg replace” → “preg_replace”
5. “Awsome-Webshell” → “Awesome-Webshell”
6. Minor sentence phrasing issues: e.g., “preg replace interpret input” should be “preg_replace interprets input…”

**Questions To Authors:**

I have a few questions. If they are already mentioned somewhere and I have missed them, please point them out. Otherwise, please let me know what you think.

1. Could a fine-tuned model outperform your prompt-based approach? Have you considered fine-tuning a model like CodeLLaMA or Qwen on your dataset?

2. How would your framework handle novel WebShells that don’t use any of the “critical” functions?

3. For real-world deployment, what setup do you recommend? Can smaller models run BFAD effectively in production? Will you release the code and dataset?

**Reasons To Accept:**

1. Timely and relevant problem: WebShell attacks are growing and demand better detection tools.
2. Novelty: This is the First comprehensive study of LLMs for this domain as far as I know
3. Strong empirical foundation: Evaluates multiple LLMs against competitive baselines with robust metrics.
4. Significant improvements: BFAD boosts F1 scores by an average of 13.8%, with top models exceeding SOTA performance.
5. Insightful analysis: Clear diagnosis of LLM weaknesses and targeted fixes are also provided
6. Clarity and transparency: Well-written with detailed methods, results, and limitations.

**Reasons To Reject:**

1. Dependence on large models: I think top results rely on expensive, proprietary models (e.g., GPT-4), which are not available to everyone
2. System complexity: BFAD includes multiple modules (filters, embedding-based selection), which might be a significant overhead
3. Limited error analysis: There is always more room for improvement and could explore false positives/negatives and failure cases in more depth.
4. Narrow scope: Only PHP-based WebShells are studied; generalizability to other scripting environments is not shown.

---

> ### Author Response · Authors · 2025-05-31
>
> We sincerely thank you for the positive evaluation and recognition of our work. We are pleased that you appreciate our empirical foundation and insightful analysis with BFAD achieving significant improvements.
>
> We would first like to address the specific questions you raised:
>
> **Q1: For real-world deployment, what setup do you recommend? Can smaller models run BFAD effectively in production? Will you release the code and dataset?**
>
> Due to resource limitations, we currently evaluate LLMs via API access. For enterprise deployment, security and data privacy are critical, so we recommend local deployment of models. Without fine-tuning, we suggest using models with at least 14B parameters to achieve competitive performance. Our core code has been included in the supplementary materials. All datasets used are publicly available, and we plan to release the full code and dataset upon paper acceptance.
>
> **Q2: Could a fine-tuned model outperform your prompt-based approach? Have you considered fine-tuning a model like CodeLLaMA or Qwen on your dataset?**
>
> As noted in *swDn (Reviewer 2) Q4*, our current work represents the first attempt to explore the use of LLMs in WebShell detection. We focused on demonstrating the promise of this approach via prompt-based methods. Given the encouraging results, we are confident that future work involving fine-tuning has the potential to achieve even better performance.
>
> Given that LLMs have not yet been widely applied to WebShell detection, we summarize current limitations and outline potential future research directions. Our goal is to help advance this emerging field and inspire more comprehensive exploration. You can also access this part in https://docs.google.com/document/d/1pMab2TGHICOvDzc2LdDox9GP2smx-YAaBM6xdlz2FHM/edit?usp=sharing
>
> **Q3: How would your framework handle novel WebShells that don’t use any of the “critical” functions?**
>
> Our framework can serve as a first-stage filter for completely novel attack vectors and can be improved by:
>
> 1. **Leveraging reasoning models**: For ambiguous or uncertain samples (e.g., using entropy as a threshold), we plan to employ current LLMs with better reasoning capabilities to emulate human analysts in interpreting unfamiliar behaviors.
> 2. **Dynamic expansion via agent workflows**: We envision integrating rulebase growth into an automated agent framework, allowing the system to detect and adapt to newly emerging patterns without manual intervention.
> 3. **Incorporating dynamic analysis**: Recent research has shown that runtime function call traces collected through sandbox execution can reveal encrypted or hidden function behaviors. We believe such dynamic data can significantly improve our framework’s ability to detect stealthy WebShells and enhance the effectiveness of our Critical Function Filter in more challenging cases. We are actively collaborating with industry partners to explore this direction.
>
> At present, we have not been able to further validate these extensions. As noted in Reviewer 1h5H (Reviewer 1) Q2 and swDn (Reviewer 2) Q6, nearly all academic work on WebShell detection relies on open-source datasets, due to privacy constraints and limited access to real-world data.
>
> That said, considering the promising results LLMs have already achieved on open-source data, we remain optimistic that advances in LLM capabilities will continue to drive progress in this direction.

---

> > ### Author Response · Authors · 2025-05-31
> >
> > **In addition to these, we would like to respond to your broader concerns:**
> >
> > **Concern: Dependence on large, proprietary models like GPT-4.**
> >
> > While our strongest results are achieved with GPT-4, BFAD also yields substantial gains on smaller models (3B–14B parameters). As open-source LLMs continue to advance and deployment costs decline, this dependency is becoming less of a constraint. BFAD is model-agnostic and compatible with any sufficiently capable LLM.
> >
> > In this study, GPT-4 was accessed through an OpenAI academic collaboration. However, as more organizations adopt self-hosted LLMs in production, we believe BFAD can be readily adapted to these settings and integrated into real-world detection systems.
> >
> > **Concern: System complexity, BFAD includes several modules that may introduce overhead.**
> >
> > While BFAD introduces multiple modules, each component addresses specific LLM limitations in code analysis. The Critical Function Filter reduces noise in long code sequences, CACE focuses attention on relevant segments, and WBFP provides behaviorally-relevant context. This modular design allows for flexible deployment where organizations can choose components based on their computational constraints.
> >
> > As noted in *1h5H (Reviewer 1) Q3*, we have analyzed the inference cost of each component, including the LLM itself. We acknowledge that the RAG-like retrieval introduces overhead. However, given the significant gains our framework enables, we believe the trade-off is justifiable in many security scenarios.
> >
> > **Concern: Limited error analysis.**
> >
> > As noted in *1h5H (Reviewer 1)*, we added a more detailed analysis of model behavior and failure cases. We also explained a counterintuitive phenomenon raised in *swDn (Reviewer 2) Q3*—that a 1.5B model may perform worse than a 0.5B one in our framework due to ICL sensitivity.
> >
> > **Concern: Narrow scope, only PHP-based WebShells were studied.**
> >
> > As noted in *1h5H (Reviewer 1) Q4*, PHP-based program account for over **70% of website** in real-world web servers, making them a highly relevant target for our initial study. Moreover, **most existing research and datasets focus on PHP**, offering a more mature ecosystem for benchmarking. That said, we agree that extending to other scripting environments is a valuable direction for future work.
> >
> > Lastly, thank you again for pointing out the minor issues in the paper. We have corrected all the typos and phrasing problems you identified.

---

> > > ### Comment · Reviewer_EuGo · 2025-06-02
> > > **Reviewer response to authors’ rebuttal**
> > >
> > > Thank you for the detailed and constructive reply.
> > >
> > > Clarifications on deployment, code release, and dataset availability, along with the plan to support smaller open-source models, satisfactorily address my concerns regarding reproducibility and practicality.
> > >
> > > While the framework still depends on medium-to-large models for top-tier accuracy and remains evaluated only on PHP, I agree that these are reasonable first-step limitations and that the paper’s contributions stand.
> > >
> > > Your explanation of how BFAD’s modular design can be tailored to different resource budgets, along with the acknowledgement of inference-cost trade-offs, satisfactorily addresses my earlier concerns about complexity.
> > >
> > > Overall, the rebuttal reinforces my positive view of the work. I will maintain my original overall score (7 = Accept) and confidence. I look forward to seeing the final version and the public release of your resources.

---

### Official Review · Reviewer_swDn · 2025-05-10

**Rating:** 6
**Confidence:** 4
**Ethics Flag:** 1

**Summary:**

This paper explores the use of Large Language Models (LLMs) for WebShell detection, presenting a new framework called Behavioral Function-Aware Detection (BFAD). The framework addresses key challenges such as context length limitations and suboptimal in-context learning (ICL) demonstration selection through three components: Critical Function Filtering, Context-Aware Code Extraction (CACE), and Weighted Behavioral Function Profiling (WBFP). The authors conduct a thorough empirical evaluation using seven LLMs and demonstrate consistent performance improvements across all scales. While the contributions are notable and the empirical results are comprehensive, several methodological and evaluative aspects could benefit from further clarification and extension.

**Questions To Authors:**

- Have you considered testing the models under zero-day or inference-only attack scenarios without ICL examples?

- Could you clarify the procedure used to select the additional non-overlapping code segments in Algorithm 1?

- What is the rationale for not including a fine-tuned LLM baseline, given that such models are commonly used for classification tasks?

- Would you consider adding an ablation study that varies the number of ICL examples or removes components like WBFP or CACE?

- Can you comment on the performance discrepancy between Qwen-2.5-1.5B and 0.5B in Table 4?

- Given that CACE alone yields strong performance improvements, how much additional value does the hybrid strategy offer?

**Reasons To Accept:**

- The paper tackles an important and relatively underexplored problem by investigating the capabilities of LLMs in detecting WebShells, a practical and impactful cybersecurity challenge.

- The proposed BFAD framework is thoughtfully designed and addresses specific LLM limitations in this domain.

- Experimental results are extensive, and the improvements observed with BFAD, particularly for smaller models, are encouraging.

- The use of behavior-weighted similarity to select ICL demonstrations is well-motivated and shows promising empirical gains.

**Reasons To Reject:**

- Generalization: The paper would be strengthened by evaluating model robustness to unseen attack types, such as zero-day or out-of-distribution samples at inference time.

- Algorithm Clarity: Algorithm 1 could be clearer, especially regarding how the “additional non-overlapping code segments” are selected, which affects reproducibility.

- Comparative Baselines: A direct comparison with fine-tuned LLMs on the classification task would provide useful context and help isolate the benefits of the prompting-based approach.

- Ablation Studies: Including ablations (e.g., varying the number of ICL examples or isolating the contribution of each BFAD module) would offer deeper insight into where the gains come from.

- Unexpected Model Behavior: In Table 4, the smaller Qwen-2.5-0.5B model achieves higher precision than the 1.5B variant, which is counterintuitive and not discussed.

- Component Evaluation: Table 5 shows that the critical region extraction performs quite well independently. It would be helpful to more clearly quantify the added value of the hybrid approach.

- Presentation: Table 4 presents key results and would be better placed in the main body for accessibility.

---

> ### Author Response · Authors · 2025-05-31
>
> We sincerely thank you for the constructive feedback. Below, we address each of the concerns you raised.
> ### **Q1: How were non-critical segments in Algorithm 1 selected?**
> We have clarified this in revised Section 2.2.
>
> The motivation here is: using only critical code segments may mislead the model (e.g., benign functions flagged as malicious). To address this, we found a simple but effective strategy: we reattach some non-critical lines after extracting key regions (see Tables 5–6, Critical vs. Hybrid). Implementation-wise, we map extracted lines to the original code, then add top-down, non-overlapping, non-critical lines within context limits.
> ### **Q2: Why not ablate ICL example count or components like WBFP/CACE?**
> **1. ICL Examples**:
> * We follow prior practice in LLM code analysis, where ICL examples are limited (often to 1) [1].
> * WebShell tends to be long, and more ICL examples reduce space for input code, potentially degrading performance. **An additional experiment confirms this: one extra ICL example decreased F1 by 4.5% on GPT-4 and 8.2% on Qwen-2.5-3B (to be added in Section 4.3).**
> * Research has shown that more examples can hurt performance due to overfitting or context saturation [2,3]. Using one example is a safe and commonly accepted choice in practice.
>
> **2. WBFP and CACE Ablations**:
>  We have already provided comprehensive ablations:
> - Table 4: No WBFP/CACE
> - Tables 5–6: CACE only
> - Tables 7–8: CACE + WBFP
>
> We treat CACE as the foundation for directing the model's attention to security-critical code, while WBFP builds upon CACE by selecting ICL examples with similar behaviors, enabling the LLM to better interpret the extracted code. **This design follows ICL best practices** that require format alignment between input and demonstrations. Since WBFP selects examples based on behavioral similarity, the test inputs should also highlight key behavioral segments—thus the use of CACE.
> ### **Q3: Why does 0.5B outperform 1.5B in Table 4?**
> Both benefit from BFAD, but the improvement differs. **We believe this stems from ICL sensitivity and have added it to Sec 4.1**: smaller LLMs (e.g., 0.5B) heavily rely on example labels, often copying them without real reasoning. Since WBFP provides accurately matched demos, 0.5B benefits more. In contrast, 1.5B models begin reasoning more independently using pretraining knowledge, which can sometimes reduce alignment with the ICL example’s label, hurting performance if reasoning is imperfect. Larger models (e.g., 14B) can better balance source code understanding with ICL use.
>
> This aligns with prior findings: [4] shows that smaller models are more label-biased, while [5] observes that LLMs rely more on pretraining than on examples. Therefore, the 0.5B model’s stronger performance doesn’t imply higher intelligence—it simply reflects that WBFP’s selected examples are well-matched to the test cases.
> ### **Q4: Why not include a fine-tuned LLM baseline?**
> Our baselines already include code-specialized LLMs (e.g., Qwen-2.5-Coder-3B/14B), with vanilla Qwen-14B outperforming larger models like LLaMA 3.1 70B and GPT-4 (Table 4). We did not fine-tune on WebShell data, as **our current goal is to explore whether off-the-shelf LLMs with domain knowledge can beat traditional models without retraining**. This aligns with practical interest in avoiding costly retraining. Our results confirm this potential, filling a gap in current practice and offering a foundation for future fine-tuned models.
> ### **Q5: Hybrid strategy vs. CACE only?**
> For smaller models, Hybrid offers minor F1 gains over CACE. Its value lies in balancing precision and recall for flexible deployment, which we’ll clarify in the revision. However, for larger models (Table 6), Hybrid improves both metrics, making it a preferable choice in practical settings where large LLMs are feasible.
> ### **Q6: Zero-day or inference-only attacks**
> As discussed in our response to Reviewer 1h5H Q2, unlike other code analysis tasks (e.g., vulnerability detection), WebShell detection research is significantly limited by privacy and data availability. **To our knowledge, nearly all existing work uses publicly available open-sourced WebShell datasets, making it difficult to simulate diverse or novel attack types.**
>
> That said, we appreciate the valuable suggestion and have added two future directions in the revised paper:
>
> 1. Zero-day simulation: Use LLMs to generate class-specific WebShell variants for benchmarking.
> 2. Fast + slow thinking: In real-world deployments facing unseen threats, our framework can serve as a fast filter (using a pre-trained LLM with domain knowledge), followed by deeper reasoning via a second-stage model (e.g., DeepSeek-R1). Though not evaluated here due to data limits, this is a promising path.
>
>  [1] https://doi.org/10.1016/j.jss.2024.112031
>
>  [2] https://arxiv.org/pdf/2501.04070
>
>  [3] https://arxiv.org/pdf/2310.00385
>
>  [4] https://arxiv.org/pdf/2305.19148
>
>  [5] https://arxiv.org/pdf/2502.13738

---

> > ### Comment · Reviewer_swDn · 2025-06-09
> > **Response to Authors**
> >
> > Thank you for your thoughtful and detailed rebuttal. Your clarifications and additional experiments have strengthened the paper. I also appreciate the anonymous dataset you provided. I have increased my score.

---

> ### Author Response · Authors · 2025-06-08
>
> Dear Reviewer,
>
> In response to your first question, we've conducted further research and small-scale experiments to filter out extreme confounding samples from our dataset. After testing the LLM's performance on these samples, we found that Vanilla GPT-4 only detected 7 out of 20 samples (accuracy below 50%). Even the trained Graph model was only able to accurately identify 12 samples. After applying our framework, we were able to detect 14 samples correctly, though 6 remained undetected due to limitations in code comprehension.
>
> Among the correctly detected cases, we observed that the LLMs’ reasoning capabilities played a crucial role. Even when explicit malicious function calls were not directly identified, the models were often able to infer the actual behavior by analyzing obfuscated fragments and reasoning about the underlying functionality.
>
> We’ve also made our dataset publicly available, and you can access it via the following anonymous link: https://anonymous.4open.science/r/LLM-for-Webshell-Detection-DB1E
>
> We appreciate your support and look forward to hearing from you.

---

### Official Review · Reviewer_1h5H · 2025-05-13

**Rating:** 6
**Confidence:** 3
**Ethics Flag:** 1

**Summary:**

this paper explores the feasibility of using llms  for detecting WebShells, which are malicious scripts injected into web servers. Authors stated that more traditional ML/DL methods suffer from data scarcity, poor generalization, and are resource-intensive, while LLMs  “could” offer a promising alternative but they face a series of technical obstacles that make its direct application unfeasible ( working with obfuscated code, very long inputs, etc ) . The authors evaluate seven LLMs on a dataset of 26.6K PHP scripts, revealing that large LLMs achieve high precision but low recall, while smaller models show the reverse.
 Based on that finding , and to address these limitations, they propose the “Behavioral Function-Aware Detection” (BFAD) framework. BFAD consists of (1) Critical Function Filter, (2) Context-Aware Code Extraction, and (3) Weighted Behavioral Function Profiling (WBFP)
All these 3 blocks basically work together to obtain a more effective in context learning scenario, which is one the key differentiating elements of the approach (for example, there seems no use of other popular techniques such as RAG)
The proposed framework , BFAD,  improves LLM F1 scores by an average of 13.82%, with GPT-4 surpassing traditional GAT-based methods. The approach enhances recall for large models and precision for small models, mitigating token limitations and reducing hallucinations.

**Questions To Authors:**

You mention challenges with LLM context windows, yet you chose WBFP-enhanced ICL over RAG-based retrieval methods. Could you clarify why RAG was not explored as an alternative? What are the practical trade-offs between BFAD and a lightweight RAG approach in this domain?

Have you considered evaluating the framework's robustness against more advanced obfuscation techniques (e.g., polymorphism, control flow flattening)

What are the computational overheads introduced by BFAD (especially WBFP) during inference? How does the latency compare to baseline LLM usage or graph-based methods, and is the framework feasible for real-time or near-real-time detection scenarios?

(maybe out of scope) how difficult would it be  adapting BFAD to WebShells written in other languages (e.g., ASP.NET, JSP) or to general web malware detection? Would the Critical Function taxonomy and WBFP weighting require retraining or manual redefinition? I think this is quite critical in terms of generalization of the approach

**Reasons To Accept:**

In general, while this paper does not propose a novel architecture or methodology, it can be seen as a very clever orchestration of existing techniques to overcome a possibly unexplored problem. Specifically*

. WebShell detection seems to be an important but underexplored cybersecurity challenge. The paper benchmarks both large (GPT-4, LLaMa 3.1 70B) and small LLMs across a realistic dataset, I think in that sense it is trying to  fill a gap in both AI and security research.

. while the problem or scenario may seem niche, there is a consistent improvement over state of the art . for example, GPT-4 with BFAD surpass a graph-based methods (GAT) at all metrics .

**Reasons To Reject:**

The proposed approach is tailored to PHP WebShells with critical function taxonomies and behavior-specific profiling. It is unclear how generalizable BFAD would be to other malware formats (e.g., JavaScript web shells, binary exploits) . Is this an approach that only work for the specific setting set on the paper, or is there a way to quantify its generalization capabilities ?

The paper reports quantitative metrics but lacks an in-depth analysis of failure cases, such as why certain benign files are misclassified or why small models hallucinate. Understanding these failure modes is essential for practical deployment in security-sensitive applications.

Webshell in general are known for obfuscation and evasive tactics, that is where its main threat resides.  In paper , we have a bit of an idealized scenario, which is understandable as at the end it needs to be framed as a ML problem. Ideally,  the paper should have evaluated adversarial robustness (e.g., obfuscated or zero-day shells). Without this, the real-world resilience of BFAD-enhanced LLMs may remain "unproven" (this point may be a bit out of scope, but I see it as quite critical in the context of security research )

Wwhile the paper provides extensive empirical results comparing large and small LLMs, it lacks a deeper analysis explaining why larger models tend to achieve high precision but low recall, while smaller models exhibit the opposite pattern. This observed trade-off is repeatedly mentioned but remains unexplained beyond surface-level speculation (e.g., prompt sensitivity, model capacity).

---

> ### Author Response · Authors · 2025-05-31
>
> Thanks!
> We are glad you recognize BFAD as a useful approach that addresses an important but underexplored cybersecurity challenge with consistent improvement over state of the art. Below, we address each of the concerns you raised.
> ### **Main Concern: Lacks an In-depth Analysis**
> **Our Response**:  Based on your feedback, we will add a dedicated "Analysis" section (Section 4.4) containing concrete examples and a thorough breakdown of misclassification patterns. We have conducted an in-depth analysis of LLM behaviors at different scales. Details are provided here: https://docs.google.com/document/d/1pMab2TGHICOvDzc2LdDox9GP2smx-YAaBM6xdlz2FHM/edit?usp=sharing
>
> Key observations:
>
> * **Smaller LLMs** tend to rely on surface-level pattern matching. For example, Qwen-0.5B flags any use of `base64_decode` as suspicious without contextual understanding. Qwen-1.5B improves slightly by noting the use of Sodium, yet still defaults to a malicious label.
> * **Larger LLMs** (e.g., LLaMA 3.1 70B) demonstrate better contextual reasoning. They examine function usage, detect the absence of concrete malicious behavior (e.g., shell commands), and even link code to safe, open-source libraries.
>
> We hypothesize this difference stems from the number of layers and attention heads, as surface patterns and deeper semantics may be captured by different heads in different layers attending to different features.
>
> This discrepancy leads to a trade-off: smaller models suffer from lower precision (frequent misclassification of benign code due to shallow heuristics), while larger models show lower recall (missing obfuscated threats due to their broader, sometimes overly lenient reasoning). **Our proposed BFAD framework mitigates these issues** by guiding smaller models to better understand function-level behaviors and helping larger models focus on critical functions. As a result, it significantly improves detection performance across scales.
> ### **Q1: Why Not Use RAG Instead of ICL + WBFP?**
> **The key challenges in this domain are:**
>
> * Accurately identifying **behavioral signals** in complex long code;
> * Understanding the **malicious intent** behind usage patterns.
>
> To address these, we use:
>
> * **CACE** for isolating critical functional segments, and
> * **WBFP-boosted ICL** to improve understanding via retrieved, task-specific examples.
>
> **Why WBFP-enhanced ICL over RAG**: While RAG excels at retrieving general knowledge, WebShell detection requires understanding behavioral patterns rather than function definitions. We found that most LLMs already possess sufficient understanding of individual functions without RAG.  Therefore, we propose WBFP to provide a task-aware retrieval mechanism that help LLMs better understand how these critical functions are used in other samples.
>
> We think the WBFP's scoring and weighting strategies will also inspire future RAG systems that go beyond textual similarity toward behavioral relevance.
> ### **Q2: Robustness to Advanced Obfuscation**
> Our dataset follows established and widely accepted WebShell collection protocols, primarily sourcing from GitHub's open-sourced datasets. As a result, it offers limited exposure to novel obfuscated code, which is often proprietary and not publicly available.
>
> We acknowledge the current limitations and plan to enhance it via dynamic analysis (e.g., runtime function tracing) to better capture obfuscated behaviors, as discussed in the Novel WebShell Family dataset [1].
> ### **Q3: Computational Overhead and Real-Time Applicability**
> **WBFP Efficiency:**
> The WBFP is executed offline. On an RTX 3090 GPU, throughput reaches ~100 samples per minute. Using multiple threads can achieve an even faster speed.
>
> **LLM Inference:**
> LLMs offer the advantage of avoiding retraining while still achieving competitive performance via domain knowledge prompts.
>
> * GPT-4: ~1–3s/sample
> * Small LLMs (0.5B–3B): ~0.5-1s/sample
>
> In practice, this is **faster** than our local graph-based detectors. While cloud APIs and local inference differ in setup, the results suggest BFAD’s feasibility for near-real-time use, especially given the increasing prevalence of enterprise-deployed LLMs.
> ### **Q4: Generalizability to Other Languages**
> Our work focuses on PHP WebShells, as PHP still dominates server-side web development, powering 74.2% of websites globally, with just 4.5% for JS [2]. Addressing this threat remains highly relevant.
>
> **BFAD’s “filter-then-detect” design is language-agnostic.** Its core logic applies broadly; We have demonstrated its efficacy in PHP and are currently building a JavaScript version to expand the Critical Function Filter. This extension will be released as part of our future work post-acceptance.
>
> \[1] Zhao Y, Lv S, Long W, et al. *Malicious webshell family dataset for webshell multi-classification research*. Visual Informatics, 2024, 8(1): 47–55.
>
> \[2] W3Techs. *Most popular server-side programming languages*. [https://w3techs.com/](https://w3techs.com/) (Accessed May 27, 2025)

---

> > ### Comment · Reviewer_1h5H · 2025-06-03
> >
> > Thanks so much for the clarification on all the points.

---

### Decision · Program_Chairs · 2025-07-08

**Decision:**

Accept

**Comment:**

This paper evaluates a suite of LLMs on the task of detecting PHP WebShell attacks, which are a major web security vulnerability. The paper finds that large models have high precision but low recall, while small models exhibit the opposite trend. Furthermore, the paper proposes a framework called BFAD which can be used to improve LLM performance substantially. The reviewers all praised the choice of an important problem and the strong experimental rigor of the work. They pointed out a lack of error analysis, which has been addressed by the authors in their response; the authors should add this to the camera-ready version of their paper. There were also some concerns raised about whether such an approach could generalize beyond WebShell attacks, but it seems that this problem is sufficiently impactful to be a valuable contribution.